# Assessment of the Effect of the Main Grain-Producing Areas Policy on China’s Food Security

**DOI:** 10.3390/foods13050654

**Published:** 2024-02-21

**Authors:** Shaohua Wang, Haixia Wu, Junjie Li, Qin Xiao, Jianping Li

**Affiliations:** State Key Laboratory of Efficient Utilization of Arid and Semi-Arid Arable Land in Northern China, The Institute of Agricultural Resources and Regional Planning, Chinese Academy of Agricultural Sciences, Beijing 100081, China; 82101211295@caas.cn (S.W.); hxia007@126.com (H.W.); lijunjie@caas.cn (J.L.); xiaoqin@caas.cn (Q.X.)

**Keywords:** agricultural development, difference-in-differences method, security effect, entropy weight method

## Abstract

Food provided a material foundation for the development of human society and was an important cornerstone for ensuring national security. The Chinese government has always attached great importance to food security, which is not only related to economic development and social stability but also to national security and self-reliance. As the core region for grain production and the supply of staple food in China, the major grain-producing areas account for 78.25% of the total national grain output, truly earning the title of China’s “granary”. Considering the establishment of 13 major grain-producing regions across the country in 2004 as a quasi-natural experiment, the impact of policies in major grain-producing regions on ensuring national food security is examined using a difference-in-differences method based on inter-provincial panel data for 30 provinces across the country from 1997 to 2020, and the mechanisms of their effects are further analyzed. The findings show that (1) the main producing-areas policy has a significant driving effect on China’s food security, with an average annual increase of 0.0351 units in the food-security index, and the impact is expanding year by year. (2) The policy of the main grain-producing provinces mainly plays a role in guaranteeing food security by expanding the scale of grain cultivation and the scale of family land management in the main grain-producing provinces, and the scale effect of grain cultivation has a more significant impact. Further adjusting and improving the policy of the main grain-producing areas and expanding the scale-driven effect of this policy are of great significance for transforming agricultural production methods and realizing a strong agricultural country.

## 1. Introduction

Food is of great importance to the national economy and people’s livelihood, and food security is an important foundation for the long-term stability of a country. Food provided a material foundation for the development of human society and was an important cornerstone for ensuring national security [1]. Due to various factors such as climate change and regional conflicts, global food security is facing formidable challenges [2]. Food security is a fundamental issue crucial to human survival. As one of the most populous countries in the world, China is taking effective actions and playing an active role in ensuring global food security. The Chinese government has always attached great importance to food security, which is not only related to economic development and social stability but also to national security and self-reliance. Improving the comprehensive efficiency of grain production is not only the core of the “agricultural powerhouse” strategy defined in the 20th National Congress of the Communist Party of China but also one of the main objectives of the Central Document No. 1 in 2023, which emphasizes the stable production and supply of food and important agricultural products. In 2022, the total grain output in China reached 686.53-million tons, an increase of 3.68-million tons compared to 2021, with a growth rate of 0.5%. This marks the seventh consecutive year that China’s total grain output has remained above 650-million tons, achieving a continuous bumper harvest. However, as the level of economic development continues to increase and consumption patterns constantly upgrade, the demand for grain in China has been increasing year by year, and the situation of a “tight balance” between supply and demand of grain will continue. At the same time, the continuous increase in grain production has put greater pressure on resources and the environment, leading to serious degradation of water and soil resources and a shift of production focus to the north and west, exacerbating the spatial mismatch of water and soil resources. In addition, factors such as the COVID-19 pandemic, natural disasters, and unstable international situations have compounded the challenges faced by food security in the new era [3].

Since 2000, the sown area of grain in China has gradually decreased, directly resulting in a decrease in grain crop production. Combined with the impact of the 2003 severe acute respiratory syndrome (in brief SARS) epidemic and severe natural disasters, the sown area of grain dropped to below 100-million hectares, and grain production was 4306.9-million tons, a decrease of about 5.8% compared to the previous year. Meanwhile, grain consumption reached 4881 million tons, resulting in a production-consumption gap of 41.47-million tons. The production of the three major grain crops was lower than the consumption, indicating an imbalance between supply and demand [4]. In response to the increasing trend of the production-demand gap in grain, at the end of 2003, the Ministry of Finance issued the “Opinions on Reforming and Improving Policies and Measures for Comprehensive Agricultural Development”, which further defined the scope of agricultural and grain main production areas based on the major indicators such as the yield of major agricultural products in various regions. Thirteen provinces (municipalities and autonomous regions), including Heilongjiang (including the Provincial Agricultural Reclamation Bureau), Jilin, Liaoning (excluding Dalian), Inner Mongolia, Hebei, Henan, Shandong (excluding Qingdao), Jiangsu, Anhui, Sichuan, Hunan, Hubei, and Jiangxi were designated as main grain-production areas. Since the implementation of the policy for grain-main production areas, these areas have gradually become the core areas of national grain production and important guarantee zones for national food security, playing a significant role in national grain production and security [5]. In 2021, the grain production in these 13 main grain-production areas reached 52,597.5-million tons, accounting for 77.03% of the total grain output in the country, an increase of 4.36% compared to 72.67% in 2004. The sown area of grain in the main production areas was 88.07-million hectares, accounting for 74.86% of the total sown area in the country, an increase of 5.59% compared to 69.27% in 2004. It can be seen that the stability of grain supply in the main production areas directly affects the overall national food security.

The international conceptual definition of food security has undergone a developmental process from a macro to a micro perspective [6]. In 1974, the UN Food and Agriculture Organization first defined food security as “we should ensure that anyone anywhere can get enough food for survival and health in the future” [7]. The definition of food security, as agreed upon at the 1996 World Food Summit, contains four aspects: food, access, use, and stability. In 2012, the FAO updated the definition of food security, completing the development from only meeting survival needs to meeting positive and healthy living and food preferences [8].

Existing studies have mainly focused on estimating the contributions of main grain-production area policies to China’s food security from the perspective of grain supply quantity, suggesting that since the establishment of main grain-production areas, China’s grain production, sown area, and yield per unit area have all shown significant growth [9,10,11,12]. However, food security is not solely dependent on grain production. Grain production and the sown area are just as important indicators of the food-security strategy. In the new era, ensuring food security requires considering multiple goals, such as quantity security, nutritional security, ecological security, and capacity security. It is difficult to comprehensively measure the overall impact of main grain-production area policies on food security by considering only a single factor. Therefore, this study aims to investigate whether the main grain-production area policies have had a positive impact on national food security from multiple perspectives, as well as the mechanisms and regional heterogeneity of their effects.

To achieve this, this study first constructs an indicator system for evaluating food security and uses the entropy method to measure the comprehensive food-security index of 30 policies implemented in China from 1997 to 2020 in order to assess the level of food security in each province. Based on this, taking the establishment of main grain-production areas in 2004 as a quasi-natural experiment, this study adopts a difference-in-differences research method to comprehensively evaluate the effects of main grain-production area policies on food security in order to meet the requirements of high-quality development of the grain industry in the new era. The main academic contributions of this study are as follows: (1) By using the entropy method, the food-security index is more accurately calculated from multiple perspectives and levels, and the differences and trends in food security levels between main grain-production areas and non-main production areas are analyzed. (2) The main grain-production area policies and the pre-existing food-security index, which measures the degree of food security, are directly incorporated into the same research framework. By employing the difference-in-differences method, a comprehensive evaluation of the effects of main grain-production area policies in ensuring food security is conducted. (3) Using a mediation effect model, the study examines the pathways through which main grain-production area policies ensure food security and comprehensively analyzes the mechanisms of the policies.

## 2. Theoretical Analysis

### 2.1. Policy Review

Food security has the characteristics of indivisibility of utility, non-competitiveness of consumption, and non-exclusiveness of benefits, making it a public good [13]. Therefore, the establishment of main grain-production areas bears the important task of ensuring China’s food security and has a special political position. Since the first policy document in 2004, almost every year’s policy document has made explicit statements regarding the implementation of relevant policies in main grain-production areas. The specific policies mainly involve two aspects: first, the central government’s policies in charge of the development of the grain industry, including producer subsidies, incentives for grain-producing counties, and the construction of large-scale commodity grain bases; second, the inclusive policies supported by the central government for agricultural industry development in the main production areas, such as various agricultural investment funds, high-standard farmland construction, and promotion of agricultural mechanization. Furthermore, in the process of grain production, the accelerated socio-economic development, urbanization, and industrialization have increased the demand for land resources, which are necessary inputs for the development of the grain industry. Therefore, the “National Overall Land Use Plan (2006–2020)” [14] issued in 2008 clearly sets constrained indicators for the amount of arable land, the protection area of basic farmland, the scale of urban and rural construction land, per-capita urban and industrial land, and the scale of newly occupied arable land for various provinces, autonomous regions, and municipalities directly under the central government. It also requires that “a portion of the land-transfer income concentrated by provinces, autonomous regions, municipalities directly under the central government, and separately planned cities for agricultural land development should be tilted towards the main grain-production areas and key areas for land development and consolidation”, which guarantees the land scale of main grain-production areas and ensures the effectiveness of various policy implementations. Figure 1 illustrates the policy framework for supporting grain-producing regions.

### 2.2. Theoretical Analysis

The establishment of main grain-production areas, as an economic activity with externalities [15], is based on the basic logic of allocating resources to regions more suitable for grain production through policies. This promotes the agglomeration of grain production, thereby absorbing more production factors and generating positive scale effects to improve labor productivity, enabling producers to obtain economies of scale [16]. It also promotes the vertical development of the social division of labor in grain production, achieving refinement and specialization in every link from planting to harvesting [17], and stimulating the emergence of new industries and market entities to drive the development of the grain industry. Ultimately, the dual effects of these mechanisms not only generate economic benefits but also ensure national food security. Existing literature has shown that this mechanism can play a role in various aspects of ensuring food security.

Firstly, from the perspective of concentrated agricultural-production factors, the establishment of main grain-production areas and the implementation of supporting policies have promoted the scale and specialization of land and grain cultivation, thereby improving the level of agricultural mechanization [18], material capital input level [19,20,21], and technological level [22,23].

Secondly, from the perspective of farmers’ income, the establishment of main production areas has increased the income of farmers. On the one hand, scale production generates economies of scale, increasing output and operating income. On the other hand, scale production promotes the refinement of social division of labor, providing more channels for farmers to increase income and improve their wage income and operating income [24,25].

Thirdly, from the perspective of the ecological environment, the establishment of main production areas not only increases output but also has a certain degree of pollution control on the agricultural ecological environment. Agricultural non-point source pollution, particularly from fertilizer, is a significant cause of ecological degradation [26]. Studies have shown that the scale and specialization of grain production resulting from the establishment of main grain-production areas have significantly reduced fertilizer non-point source pollution while increasing grain production [27,28]. Therefore, the main grain-production area policies make it possible to ensure food security from multiple perspectives. Based on the above analysis, this paper proposes the first research hypothesis:

**H1:** 
*The policy of major grain-producing regions is conducive to ensuring China’s food security.*


The stable sown area of grains forms the foundation for the continuous growth of grain production. Before the establishment of major grain-producing regions, urbanization and marketization led to the changing patterns of producing grain, and at the same time, there was a profound transformation in agricultural production modes. On the one hand, in many regions, the strong promotion of establishing development zones and setting industrial parks harms the interests of farmers to a large extent. On the other hand, compared to the descending economic benefits of grain production, cultivating cash crops is better paid, which leads to the shrinking enthusiasm among farmers for grain cultivation. As a result, the lack of sufficient incentives prompted farmers to make a shift towards growing cash crops, inducing the rising trend of “non-grainization” in the near future. One of the most important reasons for consecutive declines in grain production from 1999 to 2003 was the continuous decrease in the planted area of grains [29]. Based on this point, the most crucial target of putting forward the policy is to expand the sown area of grains. Consequently, compared to non-major grain-producing regions, major grain-producing regions face higher demands for expanding the scale of grain cultivation, especially under the responsibility of safeguarding food security. Based on this, the second research hypothesis of this paper is:

**H2:** 
*The establishment of major grain-producing regions may effect the security of China’s food through expanding the scale of grain cultivation.*


Additionally, governments in major grain-producing regions actively advanced the adjustment of agricultural planting structure, guiding farmers to specialize in the cultivation of a single grain crop, thereby achieving large-scale cultivation of grains in major producing regions. The economies of scale will be brought out by means of large-scale management of grain production in major producing regions. Based on this, the paper proposes the third research hypothesis.

**H3:** 
*The policy of major grain-producing regions may obtain economies of scale by promoting large-scale management, thereby contributing to food security.*


## 3. Methodology

### 3.1. Variable Setting

#### 3.1.1. Dependent Variable

The level of food security can be measured according to specific quantitative indicators. However, due to the dynamically and continually adjusting character of food-security assessments, a unified set of evaluation standards has not been reached within the academic community. Domestic scholars have conducted evaluations of food-security levels from various perspectives. In terms of the scope of evaluation, research primarily focuses on the national, regional, or provincial scale. Numerous scholars have constructed food-security evaluation frameworks from different angles. Regarding evaluation methods, diverse methods are available for food-security assessments, among which quantitative research is frequently used, mainly including the composite index method [30], weighted-average method [31], principal component analysis [32], and entropy method [33,34]. Following the approach conducted by Jiang and Zhu [35], the dependent variable in this study is the food-security index constructed using the entropy method.

a.Construction of the food-security indicator system.

In 2019, China released the White Paper on China’s Food Security, which emphasizes adhering to the new development concept and pursuing high-quality development. According to the theoretical framework of the White Paper and drawing on existing research literature [36,37,38], this study constructs a food-security index consisting of 15 indicators across five dimensions. The specific evaluation indicator system is shown in Table 1.

b.Construction Method of Food-Security Index

This study employs the entropy method to construct the food-security index. The entropy method calculates the weights of various indicators in the sample based on their actual values [39], effectively avoiding weight biases caused by subjective errors. Specifically, according to the food-security evaluation-indicator system constructed in Table 1, the food-security index for China is obtained by applying the following calculation formula. The specific steps are as follows:

Firstly, to eliminate the influence of dimensions and magnitudes, standardizing both positive and negative indicator data. The standardization method is as follows:(1)$$S=\sum_{i=1}^{m}\sum_{j=1}^{n}\left(Z_{ij}\cdot W_{i}\right)$$

In the Formula (1), *W_i_* represents the weight of the indicator, *Z_ij_* is the standardized value of the *j*-th indicator for the *i*-th province, where:$$\mathrm{Positive}\;\mathrm{indicators}:\;Z_{ij}=\frac{x_{ij}-\min x_{j}}{\max x_{j}-\min x_{j}}$$
$$\mathrm{Negative}\;\mathrm{indicators}:\;Z_{ij}=\frac{\max x_{j}-x_{ij}}{\max x_{j}-\min x_{j}}$$

Calculate the proportion of the *i*-th sample under the *j*-th indicator:(2)$$P_{ij}=\frac{x_{ij}}{\sum_{i=1}^{n}x_{ij}}$$

Calculate the entropy value for the *j*-th indicator: (3)$$E_{j}=-k\sum_{i=1}^{n}p_{ij}\ln(p_{ij}),\;k=-\frac{1}{\ln(n)}>0$$

Calculate information entropy redundancy:(4)$$d_{j}=1-E_{j},j=1,2,\ldots ,m$$

Calculate the weight of each indicator:(5)$$W_{j}=\frac{d_{j}}{\sum_{j=1}^{m}\left(d_{j}\right)}$$

Finally, calculate the comprehensive index:(6)$$S_{i}=\sum_{j=1}^{m}W_{j}X_{ij}^{\prime },i=1,2,3,\ldots ,n$$
where *X_ij_*′ represents the standardized data.

The value of information entropy redundancy *d_j_* influences the size of the indicator weight *W_j_.* The larger the redundancy, the more significant the contribution to the evaluation of the food-security index, resulting in a larger indicator weight. A higher comprehensive score (*S_i_*) indicates better food security for a region.

c.Calculation results of the food-security index.

The comprehensive scores were obtained through the entropy method, and the food-security index for each region from 1997 to 2020 was determined. The food-security index in most main grain-production areas is higher than that in non-grain main production areas. Among the main grain-production areas, the top five provinces with the highest food-security indexes are Jiangsu, Shandong, Jilin, Henan, and Hebei. In non-main grain-production areas, Tianjin, Zhejiang, Beijing, Shanghai, and Fujian have relatively higher food-security indexes, indicating that these provinces, although not designated as main grain-production areas based on resource endowments, contribute to food security through their economic advantages by investing more advanced technologies and production factors in the grain industry.

In addition, as shown in Table 2, this study also calculates the average food-security index for main grain-production areas and non-main grain-production areas for each year. The results show that both the food-security index for main grain-production areas and non-main grain-production areas are increasing over time. The food-security index in the major grain-producing regions is higher than in the non-major grain-producing regions, and the difference between the two has gradually widened since 2004.

#### 3.1.2. Control Variables

Referring to the approach utilized by Tang and Jiang (2023) [40], Wu et al. (2022) [41], and Jiang and Luo (2022) [42], the following control variables are selected:Per-capita Gross Domestic Product (GDP) is used to measure the level of regional economic development, representing the economic characteristics of each region and the external economic environment for ensuring food security.Urbanization rate: Urbanization may lead to the occupation of arable land and the attraction of a high-level labor force. On the other hand, it may also provide more financial support for the promotion of technological innovation in grain production. Urbanization may have both negative and positive impacts on food security.Level of primary industry development: It measures the position and scale of the primary industry in the national economy and also represents whether the natural environment is favorable for agricultural development.Proportion of grain production: It indicates the level of grain industry development in each province.Financial support for agriculture: It represents the level of government support for agriculture, calculated as the ratio of agricultural expenditure, forestry expenditure, support for underdeveloped areas, agricultural and forestry water conservancy meteorology department expenditures, comprehensive agricultural development expenditures, and support for rural production to total public expenditure in each province.Rural electricity consumption: It reflects the level of infrastructure construction in rural areas, representing the level of agricultural modernization. To address the issue of heteroscedasticity resulting from significant differences in individual data, this study logarithmically transforms rural electricity consumption.

### 3.2. Data Sources

Since China began implementing the main grain-production area policy in 2004, to ensure data continuity and availability, this study selects panel data from 1997 to 2020 for 30 regions in China (excluding the Tibet Autonomous Region). The original data for each indicator in this study comes from the “China Statistical Yearbook” [43], “China Rural Statistical Yearbook” [44], and “China Population and Employment Statistical Yearbook” [45] for each year. Missing values are supplemented using data from local statistical yearbooks, statistical bulletins on national economic and social development, and interpolation methods. In cases where there were changes in statistical caliber, the following treatments were applied. Firstly, financial support for agriculture: Before 2007, it was calculated as the sum of expenditures on agriculture, forestry, support for underdeveloped areas, agricultural and forestry water conservancy meteorology, comprehensive agricultural development expenditures, and support for rural production as a proportion of total provincial expenditures. From 2007 onwards, this indicator represents the proportion of financial expenditures for agricultural, forestry, and water affairs in local expenditures. Secondly, urbanization rate: The calculation method is the proportion of the urban population to the total population. However, due to missing data on urban permanent residents before 2005 in some statistical yearbooks, data from 1997 to 2004 is the proportion of non-agricultural population (from the “China Population Statistics Yearbook”), while data from 2005 onwards represent the proportion of the urban population to the total population. In the process of the econometric analysis, the rural electricity consumption (*elect*) is logarithmically transformed to mitigate heteroscedasticity and non-stationarity. The statistical results of the related variables are shown in Table 3.

### 3.3. Identification Strategy 

The food-security index set in the previous section is used to measure the level of food security in each province. However, the causal relationship between main grain-production area policies and food security cannot be simply explained by the size and trend of the food security index. Therefore, this study uses the establishment of the 13 main grain-production areas nationwide in 2004 as a quasi-natural experiment to analyze the policy effects. Following the method of policy evaluation, the 13 provinces covered by the main grain-production areas are set as the treatment group, and the remaining 17 provinces (excluding the Tibet Autonomous Region) are set as the control group. The intervention year is set as 2004, and the data range selected is from 1997 to 2020 to ensure the integrity of the trends before and after policy implementation. The study employs the Difference-in-Differences (DID) method, which uses three types of information: the control group before and after 2004 and the treatment group before 2004, to construct a “counterfactual” result for the treatment group after 2004. By controlling for individual and time-fixed effects, the DID method eliminates differences between the two groups that do not change over time and include random shocks from external factors. This allows for the estimation of the treatment effects of main grain-production area policies on the food-security index while minimizing interference from other factors.

### 3.4. Baseline Model Specification

To estimate the treatment effects of main grain-production area policies on the food-security index, this study constructs the following difference-in-differences regression model:(7)$$score_{it}=\beta_{0}+\beta_{1}(treat_{i}\times post_{t})+\beta_{2}X+\mu_{i}+\lambda_{t}+\varepsilon_{it}$$

In Equation (7), *i* = 1, …, 30 represents the province and *t* = 1997, …, 2020 represents the year. The dependent variable score is the food-security index. $treat_{i}$ is a dummy variable indicating whether the province is a main grain-production area (1) or not (0). $post_{t}$ is a dummy variable indicating the time of policy implementation, with a value of 1 after 2004 and 0 otherwise. *X* represents a series of control variables directly related to agricultural production and affecting the outcome variable. *μ_i_* represents individual fixed effects, and *λ_t_* represents time-fixed effects. *ε_it_* represents the error term. The interaction term $treat_{i}\times post_{t}$, recorded as *did* represents the main grain-production area policy to be evaluated. Since Equation (7) controls for both individual and time-fixed effects, the estimated coefficient of the interaction term *β* represents the treatment effect of main-grain production area policies on the food-security index after considering the potential bias caused by non-parallel trends between the treatment and control groups.

The prerequisite for the validity of the difference-in-differences estimation is the parallel trends assumption, which requires that before the policy intervention, the treatment and control groups have similar trends in the food-security index. In addition, the baseline regression reflects the average treatment effect of main grain-production area policies on the food-security index, masking dynamic differences. To address this, an event study analysis is conducted to examine the dynamic effects of the policies in different periods. This method has become a standard approach in the literature on difference-in-differences [46]. The model is constructed as follows:(8)$$score_{it}=\beta_{0}+\beta_{1}(treat_{i}\times \sum_{t=1997}^{2020}d_{t})+\beta_{2}X+\mu_{i}+\lambda_{t}+\varepsilon_{it}$$

## 4. Analysis and Results 

### 4.1. Analysis of Baseline Model Results 

This article uses the food-security index constructed above to represent the food security level of each province. Table 4 reports the estimation results of Equation (1), where Model (1) includes only the variables and individual fixed effects and time-fixed effects without control variables, while Model (2) reports the full results including all variables. The results in Table 4 show that the coefficients of the variables are significantly positive at the 1% level, indicating that the main grain-production area policies have a significant positive effect on the food-security index constructed from various indicators. This result is consistent with the findings of Fang et al. (2022) [47]. The results of Model (2) show that the establishment of main grain-production areas increases the food-security index by an average of approximately 0.0351 units per year. H1 has been validated. The reasons are as follows: Firstly, the food security index is a value between 0 and 1 and represents a comprehensive measure. Although the coefficient of the interaction term is small, a small increase represents an overall improvement in food security. Secondly, from the subsequent results, it can be seen that the policy effect has lasted for a long time, starting from 2008 and continuing to the present, contributing significantly to the level of food security in China. Therefore, the main grain-production area policies have strong statistical and economic significance for ensuring food security in China.

### 4.2. Identification Assumption Test and Dynamic Changes in Treatment Effects 

#### 4.2.1. Common Trend Assumption

Using the difference-in-differences model requires satisfying the common trend assumption, which assumes that, in the absence of policy interventions, the trends in the outcome variable for the treatment and control groups are the same. For the purpose of this study, the common trend assumption means that, under the control of a series of observable factors, the change in the food-security index for both the treatment and control groups is consistent before the implementation of the main grain-production area policies.

Based on the baseline regression results, Figure 2 shows the coefficients of the interaction terms *β*_1_ between the grouping variable and the time dummy variables *d_t_* (with a 95% confidence interval) calculated using Equation (2). This is used to test whether the regression results satisfy the common trend assumption with 2004 as the base year. As shown in Figure 2a, under the control of a series of observable factors, the food-security index for both the treatment and control groups did not show significant differences before the implementation of the main grain-production area policies. This indicates that there were no significant differences in the food-security index between the treatment and control groups before the establishment of the main grain-production areas, which satisfies the common trend assumption.

#### 4.2.2. Dynamic Changes in Treatment Effects

Figure 2b shows the dynamic changes in the treatment effects of the main grain-production area policies after implementation. From the figure, it can be seen that from 2004 to 2020, the treatment effects of the main grain-production area policies on the food-security index generally showed a trend of increasing significance. The treatment effects started to become significant and showed a gradually expanding trend.

Specifically, there are three stages. The first stage is from 2004 to 2007. As shown in Figure 2b, the treatment effects were mostly not significant in the first four years before the implementation of the main production-area policies. This can be attributed to the fact that although the main grain-production areas were designated at the end of 2003, the corresponding supporting policies were not issued simultaneously but gradually. Additionally, there was a time-lag effect due to the seasonal restrictions of agricultural production. Therefore, the policy effects were not significant during this period. The second stage is from 2009 to 2015. During this stage, the food-security index showed fluctuating increases. The reasons behind this are that the effects of the previous stage started to emerge, and new policies continued to emerge. Under the guidance of relevant policies, grain production gradually concentrated in the main grain-production areas, and factors such as scale production and specialization in grain production had an amplifying effect on the policy effects. Therefore, the policy effects were significant not only for the main grain-production areas but also for the inclusive agricultural support policies, which were more pronounced in the main production areas than in the non-main production areas. The third stage is from 2016 to 2020. In 2016, China faced the situation of “high inventory, high imports, and high costs” in the domestic grain market, with a mismatch between the upgrading of grain consumption demand and the supply of high-quality grain and oil. Agricultural supply-side structural reforms were steadily advancing. Also, 2016 was the starting year of a new round of agricultural structural adjustment in China, aiming to improve quality and efficiency, stabilize production, and increase income. Efforts were made to reduce inventory and eliminate low-efficiency production capacity by comprehensively promoting structural changes on the supply side of agriculture [48], such as reducing the area of non-advantageous areas like “Liangda wand”. In terms of total volume, in 2016, grain production decreased by 0.8%, and the grain-sown area decreased by 0.3% compared to the previous year. These were the main reasons for the decrease in the food-security index. After 2016, reforms such as the “three agricultural subsidies”, the reform of the grain collection and storage system, and adjustments to the cropping structure were fully implemented in the main production areas. The policy effects gradually became apparent, and the food-security index showed a rapid increase.

### 4.3. Robustness Tests

#### 4.3.1. Placebo Test

To eliminate the potential interference of other unobserved omitted variables on the food-security guarantee effect of establishing main grain-production areas, this study conducts a placebo test by randomly assigning the treatment and control groups. Specifically, the order of all provincial samples in the original data is randomly shuffled, and the randomized provincial samples are then merged with the processed original dataset. The interaction terms between the randomized grouping variable $treat_{i}$ and the time dummy variables $post_{t}$ are estimated using Equation (1) to complete one placebo test. This process is repeated 1000 times, resulting in 1000 estimated coefficients *did*. As shown in Figure 3, the estimated values after randomization are mostly concentrated around 0, with a range of −0.01 to 0.01, following a normal distribution. This is lower than the estimated coefficient of 0.351 in the original model, indicating that the effect of establishing main-grain production areas on food security is indeed significant and not influenced by unobserved omitted variables.

#### 4.3.2. Propensity Score Matching with Difference-in-Differences (PSM-DID)

To ensure the robustness of the regression results, this study further employs the PSM–DID method to analyze the treatment effects of establishing main grain-production areas on food security. For ease of comparison, following the approach of Zhang et al. (2019), the propensity scores of being designated as main grain-production areas for each province are predicted using the control variables [49]. Then, nearest neighbor matching is used to match the treated samples with control samples, ensuring that the treatment and control groups have minimal differences in the food-security index before the establishment of main grain-production areas, thus reducing endogeneity issues. Based on this, the net effects of main grain-production area policies on food security are identified using the DID method. Since the PSM method can maximally address the problem of observable control variable bias and the DID method can eliminate the effects of time-invariant individual effects and time-varying time effects, the combination of the two methods can better identify policy effects. The results in Table 5, Column (1), show that the estimated coefficient *did* is consistent with the baseline regression in terms of direction, magnitude, and significance level, confirming the robustness of the positive effect of establishing main-grain production areas on enhancing food security.

#### 4.3.3. Lagged Control Variables

Considering the potential endogeneity issues between the selection of all control variables and the establishment of main-grain production areas, this study includes one-period lagged control variables in the regression to examine their significance. The results in Table 5 and Column (2) are generally consistent with the baseline regression results, confirming the robustness of the baseline regression.

### 4.4. Study of Transmission Mechanisms

The previous analysis confirms the significant positive effect of main-grain production area policies on ensuring food security. However, how are the policy effects achieved? What are the intermediate mechanisms and transmission processes through which the policies promote food security? This section will analyze these questions.

#### 4.4.1. Theoretical Analysis of Transmission Mechanisms

The scale of grain cultivation is the foundation for ensuring grain production. Expanding the scale of grain cultivation and protecting and improving comprehensive grain-production capacity are the direct goals of establishing main-grain production areas. As shown in Figure 4, the change in grain-sown area is the main cause of the change in grain production. When the grain-sown area is high, grain production is also high, and when the grain-sown area was less than 100-million hectares in 2003, grain production was at a low point. Therefore, this study believes that the most important way for main-grain production area policies to ensure food security is to guarantee the scale of grain cultivation. Due to the different resource endowments of each province, this indicator is represented by the ratio of the grain-sown area to the economic crop-sown area (*rat*).

In addition to expanding the scale of grain cultivation in terms of “quantity”, the main grain-production area policies also promote the concentration of grain production, which is the “quality” aspect. This scale effect is mainly reflected in appropriately scaled land management, which leads to an increase in per-capita production and cultivation area. With the rising production costs and relatively stable prices, the income from grain cultivation for farmers has been squeezed year by year. In the face of the contradiction between the national strategy of ensuring grain security and the rational choice of small-scale farmers to shift to non-grain production, the advantages of grain cultivation with appropriate scale have become increasingly prominent. Due to data availability, this study indirectly represents the trend of expanding and concentrating land management through the ratio of arable land area to employment in the primary industry (*parea*).

#### 4.4.2. Model Specification

This section continues to use the difference-in-differences model and adopts the analysis method of mediation effects proposed by Wen and Ye (2014) to verify the above impact mechanisms [50]. The specific testing equations are as follows:(9)$$\ln M_{it}=\alpha_{0}+\alpha_{1}(treat_{i}\times post_{t})+\alpha_{2}X+\mu_{i}+\lambda_{t}+\varepsilon_{it}$$
(10)$$score_{it}=\varphi_{0}+\varphi_{1}(treat_{i}\times post_{t})+\varphi_{2}\ln M_{it}+\varphi_{3}X+\mu_{i}+\lambda_{t}+\varepsilon_{it}$$

According to the stepwise regression method for verifying mediation effects, the first step is the baseline regression, which has been completed in the previous section. The second step is Equation (9), which verifies the effect of policy implementation on the mediator variable $M_{it}$. The third step is Equation (10), which verifies the direction and magnitude of the mediator variable. However, the stepwise regression method has low power, especially when coefficients $\alpha_{1}$ and coefficients $\varphi_{1}$ have one non-significant, further verification is needed. Therefore, the more powerful Sobel test method is used to construct the Sobel statistic, which tests the mediation effect of the mediator variable [51]. The formula for the Sobel statistic is $Z={\hat{\alpha }}_{1}{\hat{\varphi }}_{2}/\sqrt{{\hat{\alpha }}_{1}^{2}S_{\varphi_{2}}^{2}+{\hat{\varphi }}_{2}^{2}S_{{\hat{\alpha }}_{1}}^{2}}$, where *S* represents the standard error of the estimated coefficient. After passing the Sobel test, the mediation effect is quantified using the method proposed by Gelbach (2016) [52]. Gelbach believes that the total effect includes both the direct effect and the indirect effect, and the formula is $\hat{\beta }=\hat{\delta }+\sum_{j}{\hat{\alpha }}_{1}{\hat{\varphi }}_{2}$, where ${\hat{\alpha }}_{1}{\hat{\varphi }}_{2}$ represents the effect explained by the mechanism, and the effect level can also be quantified using the formula ${\hat{\alpha }}_{1}{\hat{\varphi }}_{2}*100\%/\hat{\beta }$.

#### 4.4.3. Econometric Results

Table 6 shows the econometric results of the mechanism analysis. Columns (1) and (2) present the regression results of Equation (3), and Columns (3) and (4) present the regression results of Equation (4).

Firstly, grain-cultivation scale. From the results, it can be seen that the main-grain production area policies have a significantly positive effect on the ratio of the grain-sown area to the economic crop-sown area, indicating that the policies primarily achieve the goal of ensuring food security by expanding the scale of grain cultivation. In the subsequent Sobel test, the corresponding statistic passes the significance test at the 1% level. The mechanism effect is 18.4%, indicating that 18.40% of the effect of main-grain production area policies on food security can be explained by this mechanism. In contrast to non-main production areas, main production areas have made significant contributions to ensuring food security by persisting in maintaining the scale of grain cultivation under the background of rapid urbanization and industrialization and low income from grain cultivation.

Secondly, land-management scale. From the results in Table 6, it can be seen that in the two-step regression of Equations (3) and (4), only one coefficient of *ln parea* is significant. Next, a Sobel test is conducted, and the result shows that *ln parea* and the corresponding Z-statistic pass the significance test at the 5% level. The calculated effect of the mechanism is 3.3%, indicating that main grain-production area policies have achieved a certain effect in ensuring food security through the expansion of the land-management area, although the effect of the mechanism is relatively small and weak.

## 5. Conclusions and Policy Implications

This study utilizes the grain main production area policies in 2004 as a quasi-natural experiment. By using the entropy method, it constructs a food-security index based on five dimensions to measure the level of food security in China. The study evaluates the effectiveness of the main grain-production area policies in ensuring food security, explores the mechanisms through which these policies work, and examines the potential for their continued effectiveness in the future.

The main conclusions of this study can be summarized as follows. Firstly, using the entropy method, a food-security index is constructed, and the contributions of the provinces designated as main grain-production areas to food security are analyzed. The results show that these provinces have made significant contributions to food security in China, especially in non-grain main production areas. Secondly, the baseline regression results demonstrate that the main grain-production area policies have had a significant positive effect on the level of food security in China, and this effect has been increasing in recent years. Thirdly, the mechanism analysis results indicate that the main grain-production area policies primarily work through expanding the scale of grain cultivation and improving the land-management scale.

These conclusions have important implications for adjusting main grain-production area policies in the new era. Firstly, in the face of increasing uncertainties in the international situation and the rapid industrialization and urbanization domestically, the main grain-production areas play a crucial role in ensuring food security in China. It is necessary to introduce systematic policies that protect and support the provinces designated as main grain-production areas. These policies should not only address the political importance of ensuring food security but also include a range of interest compensation measures for these provinces to ensure their enthusiasm and proactiveness in safeguarding grain production. Secondly, the main grain-production area policies should focus on both quantity and quality. It is essential to prioritize the expansion of grain-cultivation scale and continue promoting the professionalization, intensification, and appropriate scale of grain production. This can be achieved through policies that support and incentivize large-scale and efficient grain production. Attention should also be given to improving the quality and safety of grain products, keeping up with the changing demands of consumers.

In conclusion, the establishment of main grain-production areas has significantly contributed to ensuring food security in China. To further enhance food security, it is necessary to develop comprehensive policies, focus on both quantity and quality, strengthen inter-provincial cooperation, and invest in agricultural technology and innovation. These measures will contribute to the sustainable development of main grain-production areas and ensure long-term food security in China.

There are still several shortcomings in the study. First, there is no clear definition of “food-security index” in academic circles. In the study, we use the inter-provincial panel data from 1997 to 2020 to synthesize the food-security index on the basis of constructing a food-security evaluation system, but there are some functional variables that cannot be included in the index system, such as the nutrient intake, the excessive microorganisms, and the incidence of heavy metal pollution. Due to the limitations imposed by data availability, the food-security index can be further improved. Second, our study concentrates on the effect of safeguarding food security in China, and the reverse side of the main grain-producing areas policy is barely analyzed. Therefore, in the future, various factors can be included in the study so as to deepen and sharpen the research of constructing a complete framework of the main grain-producing areas policy.

## Figures and Tables

**Figure 1 foods-13-00654-f001:**
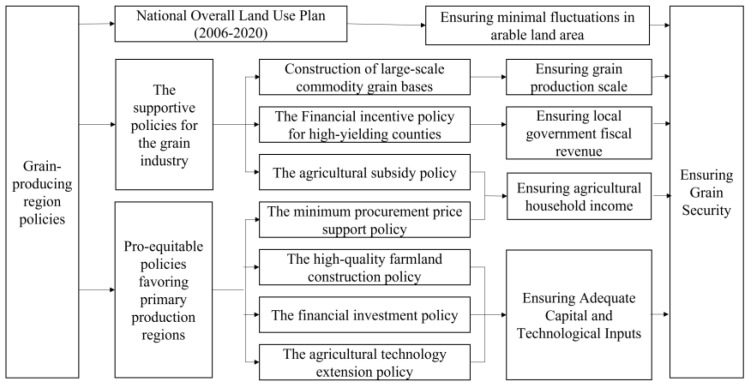
Support policy systems in major food-producing regions.

**Figure 2 foods-13-00654-f002:**
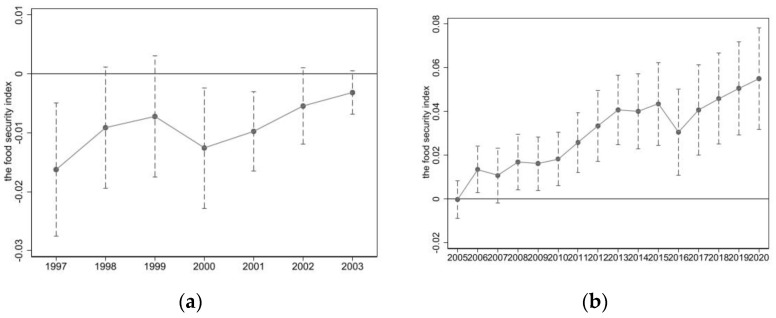
Parallel trend test and treatment affect dynamic trends. The data are based on the regression results of Equation (2), with 2004 as the base year. The dotted lines passing through the circles represent the 95% confidence interval of the estimated parameters. (**a**) represents the results from 1997 to 2003, while (**b**) illustrates the results from 2005 to 2020. The year 2004 serves as the policy implementation year and is set as the baseline year.

**Figure 3 foods-13-00654-f003:**
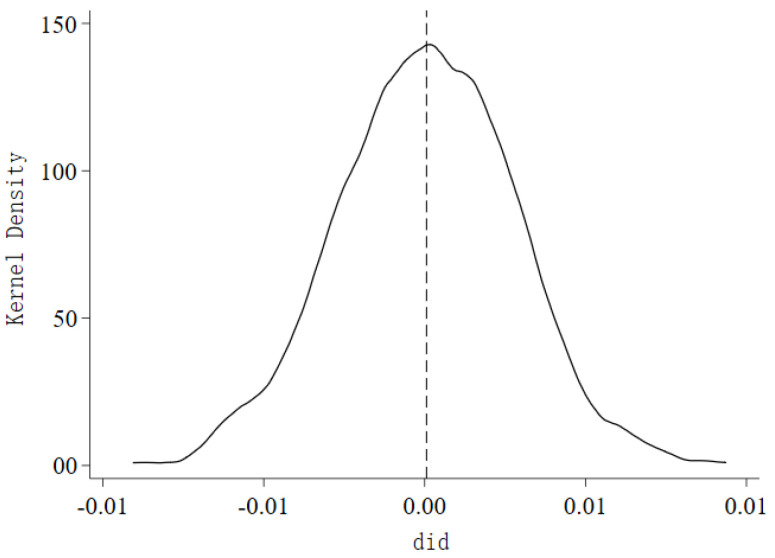
Placebo test results.

**Figure 4 foods-13-00654-f004:**
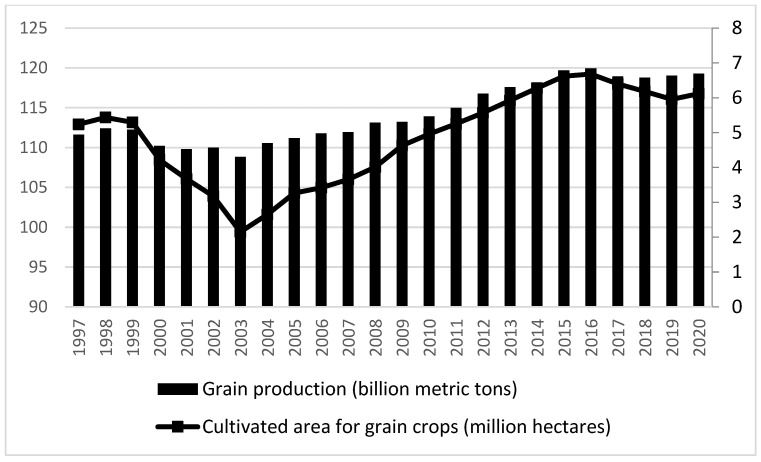
Grain production and area sown to grain.

**Table 1 foods-13-00654-t001:** Food-Security Evaluation-Indicator System in China.

Primary Indicators	Secondary Indicators	Calculation Method	Indicator Attribute
Production Capacity	Grain yield per unit area	Grain production/Grain-sown area	Positive
	Planting structure	Grain-sown area/Crop-sown area	Positive
Factor Input	Investment in farmland irrigation	Farmland effective irrigation area/Total crop-sown area Positive	Positive
	Agricultural machinery input	Total power of agricultural machinery/Total crop-sown area	Positive
	Proportion of primary industry employment	Primary industry employment/Permanent population	Positive
Producers’ Livelihood	Rural residents’ consumption level	Rural residents’ consumption level/CPI	Positive
	Operating net income	Farmers’ operating net income	Positive
Supply Stability	Grain transportation	Length of railways, highways, waterways/Provincial area	Positive
	Accessibility of grain	Per-capita grain possession Positive	Positive
	Fluctuation in grain-sown area	Previous year’s grain-sown area—Current year’s grain-sown area	Negative
	Fluctuation in total grain production	Previous year’s total grain production—Current year’s total grain production	Negative
	Fluctuation in grain prices	Previous year’s consumer price index for grain—Current year’s price index for grain	Negative
Production Sustainability	Unit area of fertilizer input	Fertilizer application/Total crop-sown area Negative	Negative
	Unit area of pesticide input	Pesticide use/Total crop-sown area	Negative
	Unit area of plastic film use	Plastic film use/Total crop-sown area	Negative

**Table 2 foods-13-00654-t002:** Mean and difference between food-security indices for main and non-main production areas, 1997–2020.

Year	Average Value of Food Security Index in Main Producing Areas	Average Value of Food Security Index in Non-Major Grain-Producing Areas	Difference
1997	0.2456	0.2403	0.0053
1998	0.2563	0.2455	0.0109
1999	0.2598	0.2453	0.0145
2000	0.2522	0.2460	0.0062
2001	0.2569	0.2464	0.0105
2002	0.2654	0.2518	0.0135
2003	0.2643	0.2518	0.0125
2004	0.2707	0.2515	0.0191
2005	0.2775	0.2592	0.0183
2006	0.3167	0.2833	0.0333
2007	0.3234	0.2902	0.0332
2008	0.3378	0.2970	0.0408
2009	0.3416	0.3034	0.0382
2010	0.3551	0.3136	0.0415
2011	0.3751	0.3241	0.0510
2012	0.3879	0.3321	0.0558
2013	0.4036	0.3412	0.0624
2014	0.4181	0.3580	0.0601
2015	0.4346	0.3721	0.0626
2016	0.4374	0.3825	0.0550
2017	0.4580	0.3947	0.0633
2018	0.4543	0.3875	0.0669
2019	0.4672	0.3984	0.0687
2020	0.4718	0.3966	0.0752

**Table 3 foods-13-00654-t003:** Results of descriptive statistics for each variable.

Varible	Name	Unit	Mean	SD	Calculation Method
*score*	Food Security Index	-	0.33	0.09	Entropy method
*pgdp*	Per Capita GDP	Ten thousand yuan	3.29	2.86	GDP/Total population
*urban*	Urbanization Rate	%	47.16	18.32	Urban population/Total population
*prim*	Level of Primary Industry Development	%	12.95	7.27	Value of primary industry/GDP
*foodpro*	Proportion of Grain Production	%	3.30	2.57	Grain production of each province/Total national grain production
*finpro*	Financial Support for Agriculture	%	10.07	3.53	Expenditure on agriculture/Total expenditure
*elect*	Rural Electricity Consumption	100-million kilowatts/hour	198.93	327.62	Annual rural electricity consumption of each province

**Table 4 foods-13-00654-t004:** Baseline regression results.

	(1)	(2)
	*Score*	*Score*
*did*	0.0392 ***	0.0351 ***
	(0.0100)	(0.0058)
*pgdp*	-	0.0118 ***
	-	(0.0019)
*urban*	-	0.0015 **
	-	(0.0006)
*prim*	-	0.0008
	-	(0.0010)
*foodpro*	-	0.0177 ***
	-	(0.0035)
*finpro*	-	0.0003
	-	(0.0007)
*elect*	-	0.0034
	-	(0.0045)
*cons*	0.2426 ***	0.1010 **
	(0.0043)	(0.0442)
*R* ^2^	0.9100	0.9430
*N*	720	720

Standard errors clustered at the province level are shown in parentheses. *** and ** represent statistical significance at the 1% and 5% levels, respectively. Model (2) in Table 4 omits the regression results for control variables and the constant term, the same below.

**Table 5 foods-13-00654-t005:** Robustness test results.

	(1)	(2)
	Nearest Neighbor Matching	Lagged One Period
*did*	0.03505 ***	0.03514 ***
	(0.0058)	(0.0062)
*control*	Yes	-
*L. control*	-	Yes
*Individual Fixed Effects*	Yes	Yes
*Time Fixed Effects*	Yes	Yes
*R* ^2^	0.9430	0.9613
*N*	720	690

Nearest Neighbor Matching uses a 1:3 matching method. Standard errors clustered at the province level are shown in parentheses. *** represent statistical significance at the 1% levels, respectively.

**Table 6 foods-13-00654-t006:** Regression results for mechanism analysis.

	(1)	(2)	(3)	(4)
	*Lnratio*	*Lnparea*	*Score*	*Score*
*did*	0.2820 ***	0.0162	0.0286 ***	0.0339 ***
	(0.0831)	(0.0196)	(0.0058)	(0.0056)
*lnratio*	-	-	0.0229 ***	
	-	-	(0.0077)	
*lnparea*	-	-		0.0709 **
	-	-		(0.0280)
*Individual Fixed Effects*	Yes	Yes	Yes	Yes
*Time-Fixed Effects*	Yes	Yes	Yes	Yes
*R* ^2^	0.9159	0.9555	0.9657	0.9645
*N*	720	720	720	720

Standard errors clustered at the province level are shown in parentheses. *** and ** represent statistical significance at the 1% and 5% levels, respectively.

## Data Availability

The data that support the findings of this study are available from the corresponding authors upon reasonable request.

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
