# Peer review of "Assessment of the Effect of the Main Grain-Producing Areas Policy on China’s Food Security"

_foods, 2024, doi:10.3390/foods13050654_

Round 1

Reviewer 1 Report

Comments and Suggestions for Authors

Review comments

This paper applied sophisticated quantitative analysis methods and various robustness tests to substantiate the author's claims. It is necessary to revise the paper considering the following points.

Major comments

1.     Before analyzing the effectiveness of the policy, an explanation or hypothesis regarding the mechanism through which this policy contributes to food security should be provided.

2.     An explanation of the specific method for calculating the Food Security Index should be provided. I am curious about how the 15 indices were aggregated to derive the Food Security Index.

3.     The food security index used in this paper differs significantly in its subcomponents from internationally recognized food security indices, such as the Global Food Security Index (GFSI), for example. It is recommended to set a commonly used and internationally comparable food security index as the dependent variable and conduct the same analysis presented in this paper.

4.     The main content of the "main grain-producing Areas Policy" revolves around investments and subsidies for agricultural infrastructure. Similarly, the key components of the food security index used in this study also represent direct government support, comprising items such as investment in farmland irrigation and agricultural machinery inputs. In other words, the constituent elements of the dependent variable, the food security index, include policy support and investments themselves. Therefore, considering the regression analysis results of this paper as indicative of policy effects may be unreasonable.

Minor comments

5.     The acronym "SARS" should be written as the full name.

6.     There is no review of prior research applying the same analytical methods to a similar topic.

Comments on the Quality of English Language

Moderate editing of English language required.

Author Response

Response to Reviewer 1 Comments

We would like to thank the reviewers for carefully reading our manuscript (Manuscript ID: foods-2845190). We appreciated the reviewers’ constructive and insightful comments very much. In the following, we include a point-by-point response to the comments from reviewer 1. In the revised manuscript, all the changes have been marked in “Track Changes” function.

Review comments and Responses

This paper applied sophisticated quantitative analysis methods and various robustness tests to substantiate the author's claims. It is necessary to revise the paper considering the following points.

Thanks for your constructive comments. All of us authors appreciate your time and encouraging words. We have carefully read the comments and revised the manuscript according to your comments.

Major comments

1.Before analyzing the effectiveness of the policy, an explanation or hypothesis regarding the mechanism through which this policy contributes to food security should be provided.

Thank you for your comments. In the revised manuscript, we have incorporated theoretical analysis and research hypotheses related to the theme of this paper. Specifically, in the theoretical analysis section (2.2 Theoretical Analysis), we have proposed three research hypotheses: Hypothesis 1 is based on benchmark regression, and Hypotheses 2 and 3 are based on mechanism analysis. We have elaborated these hypotheses to address the identified gaps in the previous discussion.

2.An explanation of the specific method for calculating the Food Security Index should be provided. I am curious about how the 15 indices were aggregated to derive the Food Security Index.

Thank you for your comments. We have incorporated the "b. Construction Method of Food Security Index" into the "3.1.1 Dependent Variable" section, where I elaborated the weight of each indice, the calculation method and the synthesis of the 15 indices.

3.The food security index used in this paper differs significantly in its subcomponents from internationally recognized food security indices, such as the Global Food Security Index (GFSI), for example. It is recommended to set a commonly used and internationally comparable food security index as the dependent variable and conduct the same analysis presented in this paper.

Thank you for your comments.Before writing the article, we consulted the internationally recognized Global Food Security Index (GFSI) and other measurement index as you mentioned. However, this paper aims to assess the policy effectiveness of the major grain-producing regions, and the implementation of this policy dated back to 2004. To measure the policy effects, a continuous, long-term panel dataset covering the period around 2004 is required for before-and-after policy comparisons. Unfortunately, the existing international food security index do not meet the given requirement.This article constructs an indicator system based on previous research results and some government documents, and incorporates as many indicators as possible into the framework. first, the food security index employed in this paper is not only inspired by the research findings written by scholars such as Wu (2019), Jiang and Zhu (2021) but also constructed based on their prior research. Second, the food security index developed in this study takes into account both basic principles such as representativeness and availability of data and guidelines of the "China's Food Security" white paper.

4.The main content of the "main grain-producing Areas Policy" revolves around investments and subsidies for agricultural infrastructure. Similarly, the key components of the food security index used in this study also represent direct government support, comprising items such as investment in farmland irrigation and agricultural machinery inputs. In other words, the constituent elements of the dependent variable, the food security index, include policy support and investments themselves. Therefore, considering the regression analysis results of this paper as indicative of policy effects may be unreasonable.

Thank you for your comments.Among the 15 indicators used in this study, only two indicators named " Investment in farmland irrigation " and " Proportion of primary industry employment," directly involve government support. The reason for incorporating these two indicators into the food security index is twofold. Firstly, to promote increased grain production and income, ensuring food security, these two policies are not only implemented in the major grain-producing regions but also nationwide. Secondly, the capacity for ensuring food security should be a multidimensional and comprehensive concept, representing an integrated capability with the new national food security strategy at its core. This paper utilizes the entropy method to synthesize these 15 indicators into a composite index, mitigating potential biases that may arise from these two indicators and explanatory variables.

Minor comments

5.The acronym "SARS" should be written as the full name.

Thank you for your comments. We have corrected "SARS" to "the 2003 severe acute respiratory syndrome (in brief SARS) epidemic." 

6.There is no review of prior research applying the same analytical methods to a similar topic.

Thank you for your comments. We have taken note of the concern regarding the lack of a review of prior research applying the similar analytical methods to a similar topic. In response, we have incorporated a comprehensive review of the research methodology in the "3.1.1 Dependent Variable" section.

Reviewer 2 Report

Comments and Suggestions for Authors

Dear Authors,

The article is good.

The correction suggestion is attached.

Best regards,

Comments on the Quality of English Language

Dear Authors,

The article is good.

The correction suggestion is attached.

Best regards,

Author Response

Response to Reviewer 2 Comments

      We would like to thank the reviewers for carefully reading our manuscript (Manuscript ID: foods-2845190). We appreciated the reviewers’ constructive and insightful comments very much. In the following, we include a point-by-point response to the comments from reviewer 2. In the revised manuscript, all the changes have been marked in “Track Changes” function.

Review comments and Responses

Thank you for your suggestion. I have made modifications to the keywords in the article, removing "major grain-producing areas" and "food security". The revised keywords are "agricultural development; differences-in-differences method; security effect; entropy weight method ".

Reviewer 3 Report

Comments and Suggestions for Authors

Having read this manuscript, I find it relevant and publishable with the following improvements.

Abstract: I suggest two things here. First, further introductory details on food security and grain production in China need to be provided. Second, justify the study prior to discussing its methodology and outcomes. Together, these would enhance comprehension of the abstract.

Introduction: The introduction starts with “China” perspectives on crop production. As this journal is international and has an international readership, I suggest that the authors begin with a more general/global introductory statement that orientates the general subject before diving into the Chinese aspect.

Structure and content: There is a little confusion on the transition from sections 3 to 5. Where does the methodology begin and end in the manuscript? I suggest that the authors strictly delineate sections 3 – 7 into three key categorical sections representing “Methodology” (Sections 3 and 4) and “Analysis and results” (sections 5-6) and end with a “Discussion and conclusion” or Conclusion and policy implications” (Section 7). This will make the entire manuscript more structured for easier comprehension. More discussion on policy should be introduced in the last section.

References: The referenced materials should be enhanced to include more international literature (remember my comments on the introduction). The current reference list reflects primarily Chinese literature.

Comments on the Quality of English Language

The language is good and requires only minor improvements.

Author Response

Response to Reviewer 3 Comments

        We would like to thank the reviewers for carefully reading our manuscript (Manuscript ID: foods-2845190). We appreciated the reviewers’ constructive and insightful comments very much. In the following, we include a point-by-point response to the comments from reviewer 3. In the revised manuscript, all the changes have been marked in “Track Changes” function.

Review comments and Responses

Comments and Suggestions for Authors

Having read this manuscript, I find it relevant and publishable with the following improvements.

Abstract: I suggest two things here. First, further introductory details on food security and grain production in China need to be provided. Second, justify the study prior to discussing its methodology and outcomes. Together, these would enhance comprehension of the abstract.

Thank you for your feedback. In the abstract section, we have incorporated the relevant content before discussing the results. We have included the background of the food security policy and data on grain production in the main grain-producing areas to provide a more comprehensive overview in the abstract.

Introduction: The introduction starts with “China” perspectives on crop production. As this journal is international and has an international readership, I suggest that the authors begin with a more general/global introductory statement that orientates the general subject before diving into the Chinese aspect.

Thank you for your suggestion, and I have incorporated relevant content into the introduction section of the paper. It positions my research from an international perspective and includes discussions on related topics in the international context.

Structure and content: There is a little confusion on the transition from sections 3 to 5. Where does the methodology begin and end in the manuscript? I suggest that the authors strictly delineate sections 3 – 7 into three key categorical sections representing “Methodology” (Sections 3 and 4) and “Analysis and results” (sections 5-6) and end with a “Discussion and conclusion” or Conclusion and policy implications” (Section 7). This will make the entire manuscript more structured for easier comprehension. More discussion on policy should be introduced in the last section.

Thank you for your suggestion. In the latest draft, I have removed the heading for Section 4 and merged its content into Section 3; Combine the original sections 5 and 6 into section 4; change the title of the original section 7 to " Conclusion and policy implications ".

References: The referenced materials should be enhanced to include more international literature (remember my comments on the introduction). The current reference list reflects primarily Chinese literature.

Thank you for your suggestion. We have added more references to international journals in the manuscript.

Reviewer 4 Report

Comments and Suggestions for Authors

In Introduction (Line 33-34), the grain production units are wrong. The unit of 13731 announced by the National Bureau of Statistics of China is not kilograms.

Line 49: China's grain production in 2003 should be 430.69 million tons.

The units of grain production mentioned in Introduction should be consistent to facilitate judgment.

Line 159-160: Chemical fertilizer pollution is mentioned as the main cause of ecological degradation. What is the scientific evidence?

The full text is expected to comprehensively consider factors affecting food security, but the evaluation factors do not take into account the safety of the four aspects of food security mentioned in Line 77-79 (the incidence of overdose, microorganisms, and heavy metal contamination) and the adequacy of nutrients, etc. The evaluation is only conducted from the perspective of sustainability and econometrics, with overemphasis on the social science aspect.

Moderate application of pesticides and chemical fertilizers can effectively increase grain production, but this study uses them directly as negative indicators, which is biased. The impact of pesticide application on yield has not been evaluated.

In Table 2, no statistical analysis was performed to identify the starting year when the food safety index of the two regions was significantly different.

In Table 3, the standard deviation is too large. How to judge the statistical reliability and validity of the data?

Line 308: The text in Table 4 should be deleted. If *(0.05<p<0.1) is not used, it does not need to be listed. In addition, the footer of each table should re-describe this labeling principle.

Line 330: Mentioning that Figure 2.(b) indicates significant growth, what is the test method used? From which year does it differ significantly from the starting year?

Line 428: it should be "Figure" 4.

In Figure 4, it is not considered whether there are significant differences in the output per unit area in each year. For example, although the planting area was greatly reduced due to SARS in 2003, the yield did not seem to decline significantly. Observing the relationship between the bar and the polyline in Figure 4, it is contradictory to the author's discussion.

Overall, there are some problems such as mistyping of data and incomplete statistical descriptions, and there are still many contradictions between the views. It is recommended to carefully revise, adjust and supplement them.

Author Response

Response to Reviewer 4 Comments

We would like to thank the reviewers for carefully reading our manuscript (Manuscript ID: foods-2845190). We appreciated the reviewers’ constructive and insightful comments very much. In the following, we include a point-by-point response to the comments from reviewer 4. In the revised manuscript, all the changes have been marked in “Track Changes” function.

Review comments and Responses

In Introduction (Line 33-34), the grain production units are wrong. The unit of 13731 announced by the National Bureau of Statistics of China is not kilograms.

Line 49: China's grain production in 2003 should be 430.69 million tons.

The units of grain production mentioned in Introduction should be consistent to facilitate judgment.

Thanks for your suggestion. We have made modifications in the corresponding positions of the manuscript.

Line 159-160: Chemical fertilizer pollution is mentioned as the main cause of ecological degradation. What is the scientific evidence?

Thanks for your suggestion. To start with, Chinese government currently has many official documents proving that the ecological environment is in an adverse situation, and the surface source pollution from fertilizers is the noticeable one. In recent years, multiple reports regarding excessive fertilizer use in major grain-producing regions are performed, so the disadvantage to the ecological environment is notable. Secondly, this result is also evident in some literature. For example, Ma et al. (2021) argue that the surface source pollution emitted from agricultural production poses a serious threat to China's ecological environment. We have added the reference in our paper to provide more scientific evidence.

The full text is expected to comprehensively consider factors affecting food security, but the evaluation factors do not take into account the safety of the four aspects of food security mentioned in Line 77-79 (the incidence of overdose, microorganisms, and heavy metal contamination) and the adequacy of nutrients, etc. The evaluation is only conducted from the perspective of sustainability and econometrics, with overemphasis on the social science aspect.

Thanks for your suggestion.When constructing the food security indicators, due to the necessity for a long-term panel dataset spanning 2004 and based on data availability, we were unable to find relevant data representing the incidence of overdose, microorganisms, and heavy metal contamination and the adequacy of nutrients.We acknowledge this limitation in our study. We will address this deficiency by incorporating this aspect in the final paragraph of the paper.

Moderate application of pesticides and chemical fertilizers can effectively increase grain production, but this study uses them directly as negative indicators, which is biased. The impact of pesticide application on yield has not been evaluated.

Thanks for your suggestion. We do realize that categorizing fertilizers and pesticides as negative indicators causes bias,but the entropy method requires defining indicators as either positive or negative for the further calculation. I designated these two indicators as negative indicators for three reasons as follows.

First, fertilizers and pesticides do have a passive impact on the ecological environment. Prolonged excessive use of fertilizers during productive process affects the physical and ecological characteristics of arable soil, leading to a series of problems such as soil compaction and a decline in the quality of the agricultural production environment, ultimately affecting grain production capacity and the safety of agricultural products. Second, there is a diminishing marginal effect of fertilizer application. Excessive use of fertilizers has been a long-standing issue. Therefore, according to the law of diminishing marginal returns, as fertilizer input continues to increase, both marginal yield and marginal output value will perform a declining trend. Third, reducing fertilizer usage has not affected food security. In China, a policy aimed at achieving zero growth in fertilizer usage was introduced in 2005. In the light of lin et al.(2024) research findings, this policy has improved the efficiency of fertilizer rather than badly impacting China's food security . Therefore, in this study, these two indicators are defined as negative indicators.

In Table 2, no statistical analysis was performed to identify the starting year when the food safety index of the two regions was significantly different.

Thanks for your suggestion. We added relevant content after Table 2.

In Table 3, the standard deviation is too large. How to judge the statistical reliability and validity of the data?

Thanks for your suggestion. Table 2 provides the descriptive statistics of the raw data, in which the standard deviation of rural electricity consumption (elect) is excessively large. Therefore, logarithmic transformation was applied to rural electricity consumption before conducting subsequent regression analysis. The detailed explanation is presented in the preceding section before Table 2.

Line 308: The text in Table 4 should be deleted. If *(0.05<p<0.1) is not used, it does not need to be listed. In addition, the footer of each table should re-describe this labeling principle.

Thanks for your suggestion. We have made the necessary corrections in the manuscript.

Line 330: Mentioning that Figure 2.(b) indicates significant growth, what is the test method used? From which year does it differ significantly from the starting year?

Thanks for your suggestion. The methodology employed in the two graphs of Figure 2 is a parallel trend test, in order to verify whether the trends before the implementation of a policy are parallel between the treatment group (those subjected to a certain policy or intervention) and the control group (those not subjected to the policy or intervention). Through this way can we ensure the net effect of the policy implement in our research. Since the base year is set as 2004, and there is no data for the year 2004, the parallel trend test is divided into two charts based on the year 2004. Chart (a) presents the results for the years 1997-2003, indicating no significant difference between the major grain-producing regions and the non-major grain-producing regions, as the confidence intervals span across value 0. Chart (b) shows the results for the years 2005-2020, indicating a significant difference between the major and non-major grain-producing regions approximately in 2006 (confidence intervals do not span value 0 ), with the regression coefficient gradually increasing.

Line 428: it should be "Figure" 4.

Thanks for your suggestion. We have made the necessary corrections in the manuscript.

In Figure 4, it is not considered whether there are significant differences in the output per unit area in each year. For example, although the planting area was greatly reduced due to SARS in 2003, the yield did not seem to decline significantly. Observing the relationship between the bar and the polyline in Figure 4, it is contradictory to the author's discussion.

Thanks for your suggestion. From 1998 to 2003, China experienced a continuous decline in grain production. In 2003, the sown area for grains was 99,410 thousand hectares, a decrease of 4,481 thousand hectares compared to 2002, representing only 95.7% of the sown area in 2002. The grain production was 43,069.5 million tons, a decline of 2,636.3 million tons compared to 2002, accounting for 94.2% of the grain production in 2002. However, the unit yield of grain crops in 2003 was 4,332 kilograms per hectare, a decrease of 67 kilograms per hectare compared to 2002, representing 98.5% of the unit yield in 2002. Therefore, as the unit yield did not show an apparent descent, so we can deduce that the main reason for the reduction in grain production was a substantial decrease in number of acres planted to grain.

Overall, there are some problems such as mistyping of data and incomplete statistical descriptions, and there are still many contradictions between the views. It is recommended to carefully revise, adjust and supplement them.

Thanks for your suggestion. We will carefully review and make the necessary revisions.

Reviewer 5 Report

Comments and Suggestions for Authors

Peer-reviewed article entitled “Assessment of the effect of the main grain-producing Areas Policy on China's Food Security” concerns a very important issue. The authors correctly justified the purposefulness of the research. The research concept is clear.

The research results confirm the effectiveness of the policy in increasing food security. Although the analyzes are correct and appropriate methods were used, similar conclusions can be reached using much less advanced research methods. The research results are quite obvious and that is why I believe that this study does not bring many new values to science. It provides reliable confirmation of known relationships. It would be much more interesting to take into account qualitative factors as well as the negative effects of the applied policy.

Despite these critical assessments regarding the originality of the work, I believe that the study should be published. This is a reliable analysis - correct in terms of content. However, it is necessary to compare the obtained results with other scientific works. The authors did not discuss the research results in the manuscript. The study would improve in quality if the authors included maps showing the location of the research area. It is also advisable to present the volume of cereal production on the maps.

Please also indicate research limitations and recommendations for future research.

Author Response

Response to Reviewer 5 Comments

We would like to thank the reviewers for carefully reading our manuscript (Manuscript ID: foods-2845190). We appreciated the reviewers’ constructive and insightful comments very much. In the following, we include a point-by-point response to the comments from reviewer 5. In the revised manuscript, all the changes have been marked in “Track Changes” function.

Review comments

Peer-reviewed article entitled “Assessment of the effect of the main grain-producing Areas Policy on China's Food Security” concerns a very important issue. The authors correctly justified the purposefulness of the research. The research concept is clear.

The research results confirm the effectiveness of the policy in increasing food security. Although the analyzes are correct and appropriate methods were used, similar conclusions can be reached using much less advanced research methods. The research results are quite obvious and that is why I believe that this study does not bring many new values to science. It provides reliable confirmation of known relationships. It would be much more interesting to take into account qualitative factors as well as the negative effects of the applied policy.

Despite these critical assessments regarding the originality of the work, I believe that the study should be published. This is a reliable analysis - correct in terms of content. However, it is necessary to compare the obtained results with other scientific works. The authors did not discuss the research results in the manuscript. The study would improve in quality if the authors included maps showing the location of the research area. It is also advisable to present the volume of cereal production on the maps.

Please also indicate research limitations and recommendations for future research

Responses:

Point 1: The authors did not discuss the research results in the manuscript.

Thank you for your comments. We have added a discussion of the results at the relevant section in the article.

Point 2:

The study would improve in quality if the authors included maps showing the location of the research area. It is also advisable to present the volume of cereal production on the maps.

Thank you for your suggestion. Adding a map and annotating cereal production on it could indeed enhance the quality of the article. However, due to the revision of regional boundaries in China in 2023, obtaining approval from the Ministry of Natural Resources for the use of such a map requires more time than we currently have.

Point3: Please also indicate research limitations and recommendations for future research

Thank you very much for your suggestions. We have added a section in the final paragraph addressing the limitations of the study and providing prospects for future research.

Round 2

Reviewer 1 Report

Comments and Suggestions for Authors

It is necessary to more clearly describe the limitations of this research in the conclusion part.

Comments on the Quality of English Language

Some minor English editing is needed. 

Reviewer 3 Report

Comments and Suggestions for Authors

I have gone through the revisions and I consider the current version publishable with only one amendment - that is to caption section 4 "Empirical Analysis and Results."

This will help to differentiate that analysis from section 2 which is "Theoretical analysis."

I believe that this can be done during proofing, so I will accept the manuscript for publication.

Comments on the Quality of English Language

Only minor corrections needed.

Reviewer 4 Report

Comments and Suggestions for Authors

The authors have carefully revised the manuscript.

No further question.